# Are there sex differences in physiological parameters and reaction time responses to overload in firefighters?

Fabrizio Perroni[1], Ludovica Cardinali[2], Lamberto Cignitti[3], Erica Gobbi[1], Federico Grugni[3], Stefano Amatori[4], Marco Bruno Luigi Rocchi[4], Luca Grandinetti[3], Francesco Lunetta[3], Vilberto Stocchi[1], Carlo Baldari[5], Davide Sisti[4], Laura Guidetti[2,6]*

1 Department of Biomolecular Sciences, Section of Exercise and Health Sciences, University of Urbino Carlo Bo, Urbino, Italy, 2 Department of Movement, Human and Health Sciences, University of Rome "Foro Italico", Rome, Italy, 3 Italian Firefighting Corp, Rome, Italy, 4 Department of Biomolecular Sciences, Service of Biostatistics, University of Urbino Carlo Bo, Urbino, Italy, 5 Faculty of Psychology, eCampus University, Novedrate, Como, Italy, 6 Department Unicusano, University "Niccolò Cusano", Rome, Italy

* laura.guidetti@uniroma4.it

**Data Availability Statement:** All relevant data are within the paper.

**Funding:** This research received no external funding.

## Abstract

Male and female firefighters work side-by-side in the same in strenuous and risky conditions. Anthropometrics, physiological, and reaction time (mean of reaction time -MRT-, and errors made -E) parameters of 12 Female and 13 Male firefighters were compared. Effect of overload (step test with and without equipment) on the MRT and E were analyzed on 3 trials (T1 = 1-1s, T2 = 0.5-1s, T3 = 0.5–0.5s), compared with a pre-test condition (basal). T-test between males and females was applied to assess differences (p<0.05) in all parameters. ANOVA with repeated measures and Bonferroni on 3 conditions of step test between males and females was applied in reaction time variables. Between MRT and E, in T1, T2 and T3 trials and the 3 test conditions, ANCOVA models with interactions were used. Differences (p<0.05) in anthropometric, physiological and reaction time data emerged across groups, and on the 3rd trials (T3 vs T1 and T2) in reaction time parameters of each group. ANCOVA showed differences (p<0.001) in E among trials. Post hoc showed significant differences in T1vsT3 and T1vsT2. MRT x trial interaction was extremely significant (P<0.001). Implementing fitness and reaction time exercise programs is important to decrease the injury risk and increase work capacity in firefighters with reference to female workers.

## 1. Introduction

Firefighting is a hazardous civilian occupation which is responsible for the safety of the public. It is characterized by different and variable (i.e. space, time, duration, breaks, type of the emergency) working conditions, in whose firefighters have to maintain full readiness for 24 hours (divided in shift work) and they are subjected to heavy physical demands and high psychological stress, often working under emergency circumstances [1]. In addition to the dangerous nature of the job, firefighters wear weighty and bulky Protective clothing (PC) and a Self-

**Competing interests:** The authors have declared that no competing interests exist.

Contained Breathing Apparatus (SCBA), a good barrier to avoid injuries. However, these may have negative effects on metabolic and thermal efficiency and fatigue [2–6]. Firefighting activities require optimal levels of power, strength, muscular, and anaerobic/aerobic endurance [7,8]. Various studies [8–11] reported that job performance is positively correlated with high fitness levels; the latter also showed a negative association with injury risk. Maximum Oxygen Uptake ($VO_{2max}$) and muscle strength are the most widely used measurements to assess fitness levels. It has been previously reported [11,12] that a level of 45 mL·kg$^{-1}$·min$^{-1}$ $VO_{2max}$ is required to complete severe firefighting tasks. Stevenson et al. [13] showed performance standards for the seated shoulder press test (35 kg), the seated maximal single rope pull-down test (60 kg), and the seated repeated rope pull-down test (23 reps with 28 kg), chosen as specific tests for the firefighting tasks. In addition, Perroni et al. [14] showed differences in strength-requiring trials among different firefighters age groups. Considering that firefighters must quickly adapt to the hazards present in the working environment [15], which require concentration, coordination, change and control of instinct, a higher level of cognitive functions play an important role [16]. A study of Rodrigues et al. [17] on Portuguese firefighters suggested that stress compromised cognitive performance and caused a measurable change in autonomic balance.

Despite in the United States the percentage of female firefighters increased from 1% in the 1980s to 7% in 2014 [18,19], firefighting still has one of the lowest rates of female employment for physically demanding occupations [20,21]. World fire statistics [22] estimated that female firefighters represent roughly 6.3% of firefighters; in Italy, female firefighters represent the 2.91%, and they complete the same tasks as men, having to meet the same fitness standards [23]. US Reports [24] estimates that female firefighters experienced an average of 1.260 (4%) by 30.290 injuries suffered by all firefighters from 2010 to 2014. Previously, Liao et al. [25] explored the correlation between firefighter injury frequency and duration, demographics, personality and economics, and they found that female firefighters reported 33% more injuries than their male counterparts. According to the 2013 ACSM guidelines, studies of Gendron et al. [26,27] showed that 11% of female and 34.5% of male firefighters were categorized as having moderate cardiovascular disease (CVD) risk, while 65% of female and 43.6% of male as high CVD risk.

The main goals of the present study were to analyze: 1) the fitness parameters of Italian Firefighters, comparing males and females, 2) the impact of overload, represented by the protective garments, on the reaction time to the stimulus. Despite male and female firefighters work side by side in the same, highly hard, strenuous, and risky conditions, we hypothesized that there are differences in anthropometric and fitness values between male and female Italian firefighter recruits. This study could give useful information to develop appropriate conditioning programmes aimed to reach an optimal job performance of firefighters. In fact, while much has been published on fitness requirements among male firefighters, data on female firefighters are few.

## 2. Materials and methods

### 2.1. Participants

Twenty-five healthy recruits of National Italian Firefighters Corp (Age: 37 ± 4 years; Weight: 69.1 ± 12.2 kg; Height: 169 ± 7 cm; BMI: 23.7 ± 2.8 kg/m$^2$) consisting of 12 female firefighters (FF) and 13 male firefighters (MF) were recruited to participate in this study. To avoid possible statistical differences derived by different activity levels and dietary habits, they were engaged at the end of the residential Italian Fire Fighter Corp training course, organized by Minister of Internal Affair in its public structures to became professional firefighters. The Fire Fighter

Corp training course consisted of 6 months (8 h/day, 5 days/week) of firefighting education and training, which included theoretical and practical firefighting skills (i.e., emergency medical procedures, fire apparatus operations, fire prevention, and communication practices) and strength and conditioning training. Before the course, the Italian Fire Fighting Corp selected recruits according to questionnaires, interviews, and fitness tests.

All subjects signed a consent form after a verbal and written explanation of the experimental design of this study, which was performed in accordance with the Declaration of Helsinki of the World Medical Association. The Institutional Review Board of the University of Urbino reviewed the study protocol and concluded that, since the study wasn't 'ethically sensitive' (no personal information was collected, and participants were not subjected to any risk, as they were asked to replicate tasks that they daily complete at work), the approval by an Ethics Committee was not mandatory. Verbal permission to conduct the study has been received, without any approval number/code.

## 2.2. Procedures

The testing timeline of the study is presented in Fig 1.

The participants performed two experimental sessions, at the end of the Italian Firefighters Training Course (November 2018), separated by 7 days. To minimize any order or learning effects, the experimental sessions were conducted in a randomized order. The first session was designed to determine the firefighter's anthropometric (weight, height, and body mass index) and basal fitness (Reaction Time, $VO_{2max}$ without PC & SCBA, Upper and Lower body strength, Handgrip) parameters of each subject. Immediately after the step test, used to evaluate the $VO_{2max}$, the firefighters performed a Reaction Time (RT) test to assess the effects of overload on cognitive functions. In addition, a second session was designed to perform a step test wearing the standard protective firefighting turnout gear (total weight of ~23 kg). This included a helmet, Nomex flash hood, respirator, SCBA, a cotton T-shirt, pants, boots, and gloves. Immediately at the end of the Step test, the firefighters performed the RT test again.

## 2.3. Measurements

**2.3.1. Anthropometrics measurement.** Height (cm) and weight (kg) were measured using an electronic scale (± 0.1 kg) and a fixed stadiometer (± 0.1 cm) (Seca 702, Seca GmbH

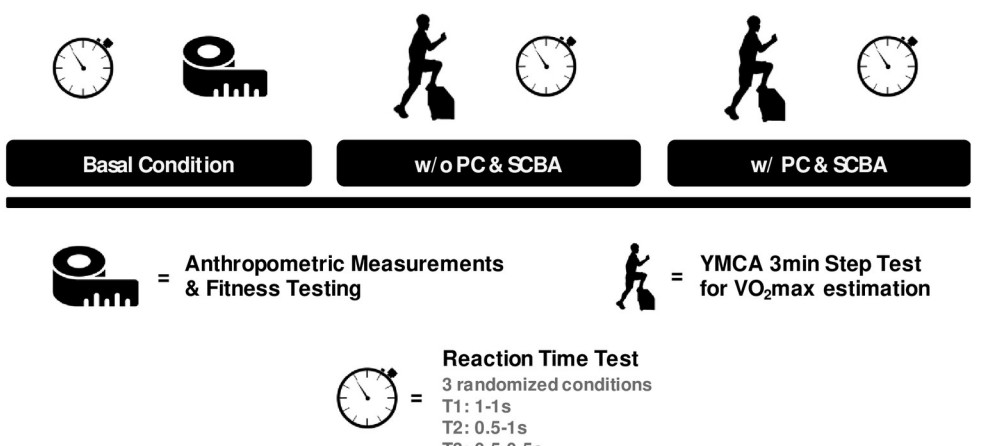

**Fig 1. Testing timeline of the study.** Reaction time tests were performed in three conditions: Basal (pre-exercise), and after the step test with and without PC and SCBA. The post step test conditions were performed in a randomized order.

& Co. KG, Hamburg, Germany), with participants barefoot and wearing light clothes. The Body Mass Index (BMI) was then calculated ($kg/m^2$).

**2.3.2. Reaction time test.** The Fitlight Trainer [R] (FitLight Sports Corp, Ontario, Canada) was used to assed the RT to visual stimulus. It is a wireless reaction system, comprised of eight LED lights controlled by a tablet computer; a sensor reacts to proximity or touch and switches off the light. Previous study of Reigal et al. [28] found adequate reliability indices of the Fitlight Trainer [R] for simple ($ICC_{2;1}$ = 0.92, SEM = 39.87, and MD = 110.51 ms) and complex ($ICC_{2;1}$ = 0.85, SEM = 63.50, and MD = 176.00 ms) task. Participants, standing upright with arms along the hips, at about 30 cm from the wall, centred to the pattern, with joined feet, had to respond as quickly as possible to the stimulus touching the green light that appeared on the disk in random order. The positioning of the light is presented in Fig 2.

After each stimulus, the subject returned to the initial position waiting for the next signal. To evaluated mean reaction time (MRT) and errors made by each subject (he/she was not quick enough to touch before the disc light deactivated; E), we purposed a test that consisted of 3 trials of 30 beats each. Trials were different for both the duration and frequency of the stimuli [29]. In particular, the three trials were defined as follows:

- T1 = 1 sec of stimulus duration and 1-sec interval between stimuli.

- T2 = 0.5 sec of stimulus duration and 1-sec interval between stimuli.

- T3 = 0.5 sec of stimulus duration and 0.5-sec interval between stimuli.

To investigate the effects of overload on RT responses, we proposed two different conditions of execution of the test: step test performed without PC & SCBA and step test performed with PC & SCBA.

**2.3.3. Step test.** We used a specific field step test compared to non-specific test (i.e. running on a treadmill) because it is a functional activity of the firefighters as climbing stairs [30].

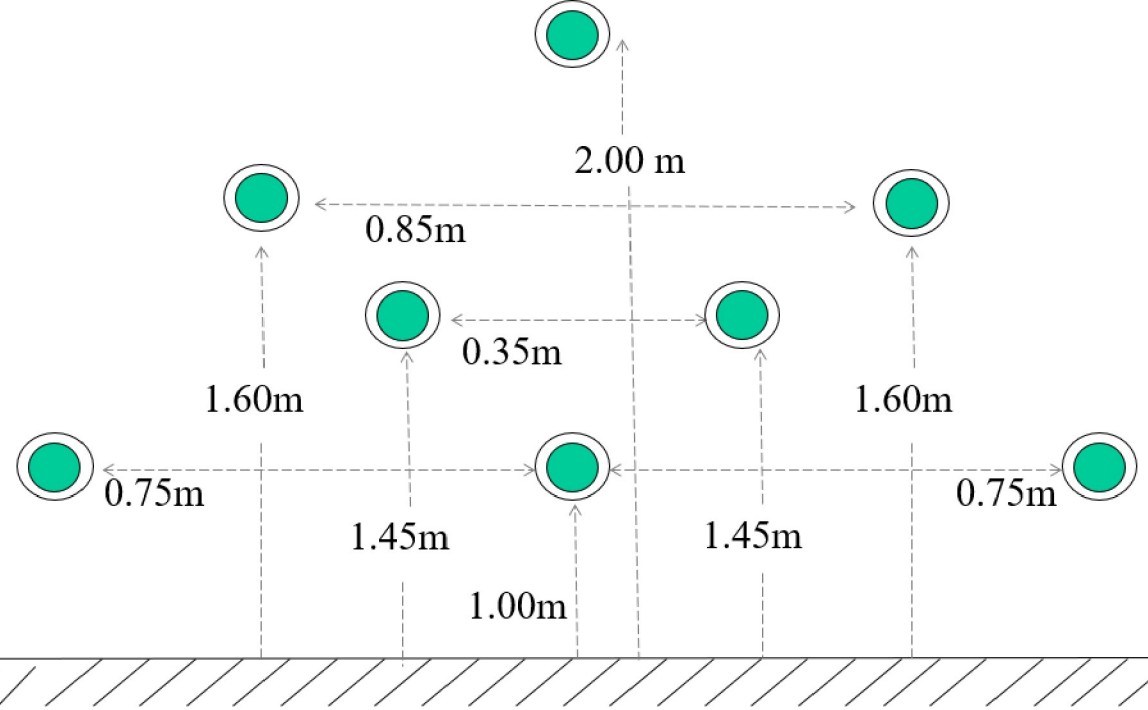

**Fig 2. Fitlight Trainer [R] positioning.**

We adopted the YMCA step test [31] consisting of stepping up and down a 12-inch step with metronome-based step frequency (96 beats and 24 step cycles per minute) for a total of 3 min, which predicts $VO_2peak$ with r = 0.83. Heart rate was continuously recorded from two min before to five min after the test, using a POLAR chest belt device (RS 400, POLAR Electro, Kempele, Finland), set to store heart rate in 5 s intervals. Subjects stayed in a temperature- and humidity-controlled quiet room during the 2 min before the test and during a 5 min recovery phase. The test was not considered completed if the participant terminated the test prematurely or did not maintain the rhythm for the 3 min of the test. After the test, we calculated the mean of twelve consecutive heart rate records in 5 s-intervals, starting 5 s after exercise termination, and we used linear regression analysis of Beutner et al. [32] for YMCA model (r = 0.86–0.91).

**2.3.4. Upper and lower body strength.** The handgrip and bench-press tests were used to evaluate the strength of the upper limbs, while an explosive power test was used to assess the strength of the lower extremities.

Dynamometer with a load-cell sensor (Lode, Groningen, The Netherlands) was used to measure the maximal peak grip strength. Each participant stood upright with arms along the hips with the dynamometer into his hand and was asked to make the strongest grip possible. Each subject performed three trials for each hand (1 min of recovery between trials), and the highest value was used for further analysis. A review of Cronin et al. [33] showed an ICC of 0.90–0.99 for the handgrip test.

To estimate the upper limb strength, subjects performed a test on a standard bench press station using a York Olympic standard barbell and free weights. For the measurement of muscle strength, a linear encoder was connected to the weight stack of the York Olympic standard barbell. After a standardized warm-up (10 submaximal repetitions at 70% of personal hypothesized one repetition maximum - 1RM), participants performed one trial, with maximum effort, consisting of four series, performed with four different weight levels, of four repetitions with 2 min of rest in between. The first external testing load was the warm-up weight and, then, the weight was increased by 2.5 kg for each weight level. On a given signal, the subject was asked to flex his or her elbow as quickly and, then, to extend it as forcefully as possible. The distance the standard barbell was lifted, and the time it took to fully flex the elbow was measured with a linear encoder connected to the barbell. The highest values performed in each series were taken for the analyses by a computerized system unit, which calculated the maximal strength in real-time (Muscle Lab, Ergotest Innovations, Stathelle, Norway). In addition, to evaluate the actual 1 RM, we used the formula of Bosquet et al. [34], which has shown that the linear encoder produced highly valid and reliable measurements in bench press testing (r = 0.93).

To measure the explosive power of the lower extremities of firefighters through a Counter Movement Jump (CMJ), an optical acquisition system (Optojump, Microgate, Udine, Italy) was used, which measures flying and ground contact times with a $10^{-3}$ s precision. Slinde et al. [35] reported that CMJ has a high test-retest stability coefficient (range 0.80–0.98). The Optojump photocells (placed 6 mm from the ground) are activated by the feet at the instance of taking-off and contact upon landing. Firefighters performed a CMJ test (stretch-shortening cycle) according to the protocol described by Bosco et al. [36] and, then, calculations of the height of the jump was calculated in real-time by a specific software [37,38]. Researchers asked to the firefighters, from the standing position, to bend their knees quickly to a freely chosen angle (around 90°) trying to avoid any knee or trunk countermovement and, immediately after, to perform a maximal vertical jump. Jumps were filmed by a video-camera and repeated if not performed correctly. To avoid possible arm-swing, firefighters had to maintain the hands on their hips, to keep their body vertical and the knees fully extended during the jump. Each subject performed three correct jumps, and the highest was taken for further analysis.

## 2.4. Statistical analyses

Descriptive statistics (means and standard deviations) were calculated to provide the anthropometric and physical fitness profile for each measured parameter. Before using statistical test procedures, the assumptions of normality and sphericity were verified by Kolmogorov-Smirnov and Mauchly's test, respectively. To check differences between groups (males vs. females), an exploratory unpaired Student's t-test was applied to all these parameters: weight, height, BMI, handgrip, bench-press, CMJ, $VO_{2max}$.

For each variable of RT test (MRT and E), analysis of variance with repeated measures on 3 conditions (Basal, After Step test performed with PC & SCBA, After Step test performed without PC & SCBA) between 2 factors (Male vs. Female) was applied. When a significant effect was found, post-hoc Bonferroni corrections were used. Finally, to test the correlation between MRT and E of subjects, in T1, T2, and T3 trials and three conditions of execution of the test, an ANCOVA model with interactions was used. In this model, E was the quantitative dependent variable, MRT was the independent quantitative one, and test conditions (3 levels) and trial type (3 levels) were the predictive categorial variables.

Heart rate time-dependent decrement (frequency measurement = 0.2 Hz) was modelized using a linear regression model; intercept (a) was the heart rate estimated at time = 0 (at the end of the session), slope (b) was average HR decrement estimated/sec and R (Pearson coefficient) was used as an index of the goodness of fitting of the linear model. Sex (between factor) and test conditions (with or without equipment; repeated measure) were predictive binary categorical variables in mixed model MANOVA, where intercept (a) and b (slope) were dependent variables. To provide meaningful analysis for comparisons between groups, eta squared effect sizes were calculated, and values of 0.1-small, 0.6-medium, 1.4-large effect were considered [39].

IBM SPSS (Ver. 23) software for Windows was used for data analysis, and significance was set at p≤0.05. GraphPad Prizm 8 was used to build the figures.

# 3. Results

The results of anthropometric and fitness data (Bench press 1RM, CMJ, Handgrip, and $VO_{2max}$) between males and females are shown in Table 1.

The heart rate, in the interval of time considered (one minute after the end of the session), followed a linear trend in all participants; Pearson's R has an average value of -0.979 (SD = 0.018), so a straight line modeled the trend almost perfectly. In Fig 3 (left panel), a single series of measurements of HR is shown as an example, with the relative regression straight line and R value. As expected, regardless of sex, the presence of the equipment showed higher values ($F_{(1, 23)}$ = 148.9, $P < 0.001$, $\eta^2$ = 0.87) of heart rate at time = 0 (at the end of the session) (intercept values: 169.6 ± 10.4), respect to the condition of equipment absence (145.3 ± 12.3), with an arise of 16.7%. Considering HR decrement estimated/sec (slope regression values), regardless of sex, the presence of the equipment showed a lower values ($F_{(1, 23)}$ = 6.10, $P = 0.020$, $\eta^2$ = 0.21) of slope (-0.56 ± 0.10) than the condition of equipment absence (-0.63 ± 0.20), with an arise of 12.5%. In brief, in both group, equipment presence caused a higher initial heart rate, followed by a slower recovery. Time x sex interaction was significant only for slopes ($F_{(1, 23)}$ = 7.25, $P = 0.013$, $\eta^2$ = 0.24) but not for intercept ($F_{(1, 23)}$ = 0.07, $P = 0.793$, $\eta^2$ = 0.003); males showed the same decrement trend, whilst females showed a higher recovery capacity, only without equipment (Fig 3, right panel); in fact, males with and without equipment showed an heart rate decrease (considering entire time recovery) respectively of -18.6% and -21.5%, whilst females of -28.2% (without equipment) and -18.2% (with equipment).

**Table 1. Means, standard deviations, and statistical differences (p < 0.05) of anthropometric and fitness data across age groups.**

| | Female (n = 12) | Male (n = 13) | t | p(t) | Total (n = 25) |
|---|---|---|---|---|---|
| Age (yrs) | 37 ± 4 | 37.1 ± 5 | 0.05 | 0.96 | 37 ± 4 |
| Weight (kg) | 59.2 ± 4.6 | 78.3 ± 9.4 | 6.36 | <0.001 | 69.1 ± 12.2 |
| Weight w/ PC & SCBA (kg) | 81.6 ± 5.0 | 101.4 ± 9.5 | 6.43 | <0.001 | 91.9 ± 12.6 |
| Height (m) | 1.64 ± 0.05 | 1.75 ± 0.04 | 6.09 | <0.001 | 1.70 ± 0.07 |
| BMI (kg/m$^2$) | 21.9 ± 2.1 | 25.5 ± 2.2 | 4.18 | <0.001 | 23.8 ± 2.8 |
| Estimated Bench Press 1RM (kg) | 34.9 ± 4.9 | 60.7 ± 10.2 | 7.95 | <0.001 | 48.3 ± 15.3 |
| Measured Bench Press 1RM (kg) | 39.9 ± 4.9 | 66.2 ± 10.4 | 7.95 | <0.001 | 53.6 ± 15.6 |
| Countermovement Jump (cm) | 23.1 ± 2.5 | 28.8 ± 4.1 | 4.15 | <0.001 | 26.0 ± 4.4 |
| Handgrip (kg) | 31.5 ± 2.9 | 48.1 ± 11.4 | 4.89 | <0.001 | 40.1 ± 11.9 |
| VO$_{2max}$ (ml·kg$^{-1}$·min$^{-1}$) | 45.6 ± 3.1 | 41.5 ± 2.6 | 3.59 | <0.01 | 43.5 ± 3.5 |
| VO$_{2max}$ (L·min$^{-1}$) | 2.7 ± 0.2 | 3.3 ± 0.5 | 4.68 | <0.01 | 2.9 ± 0.5 |
| VO$_{2max}$ w/ PC & SCBA (ml·kg$^{-1}$·min$^{-1}$) | 41.0 ± 2.5 | 37.8 ± 2.2 | 3.40 | <0.01 | 39.3 ± 2.8 |
| VO$_{2max}$ w/ PC & SCBA (L·min$^{-1}$) | 3.3 ± 0.3 | 3.8 ± 0.4 | 3.51 | <0.01 | 3.6 ± 0.4 |

For the variables of the RT test, a baseline score reports the average of the pre-test scores measured before the three physical exercise sessions. Results of MRT and E between males and females are shown in Fig 4. MRT was significantly associated only with trials (p<0.05) but not with different test conditions and sex (p>0.05); this means that, regardless of the test condition, MRT depended only on trial. Post hoc analysis showed that T1 condition was different from T2 and T3 conditions. Finally, E, as MRT, was significantly associated only with trials (p<0.05) but not to different test conditions; post hoc analysis showed that T1 condition was different from T2 and T3 conditions. In addition, a significant interaction (sex x MRT; p<0.05) was found; the number of errors was slightly higher in females, mostly in w/ PC & SCBA condition.

A two-way ANCOVA was conducted to analyze the effect of trials and test conditions while controlling for MRT. There was a significant difference in E ($F_{2,215}$ = 110, p<0.001) among trials. Post hoc tests showed that there were significant differences between trials 1 and 3 (p<0.001) and trials 1 and 2 (p<0.001), while no significant difference (p>0.05) was present between trial 2 and 3. No significant differences in E ($F_{(2,215)}$ = 0.438, p>0.05) among test conditions were found. This means that regardless of the trial, there were, on average, no differences among basal, after step test performed with PC & SCBA, and after step test performed without PC & SCBA. MRT x test execution interaction was not significant, and slopes among

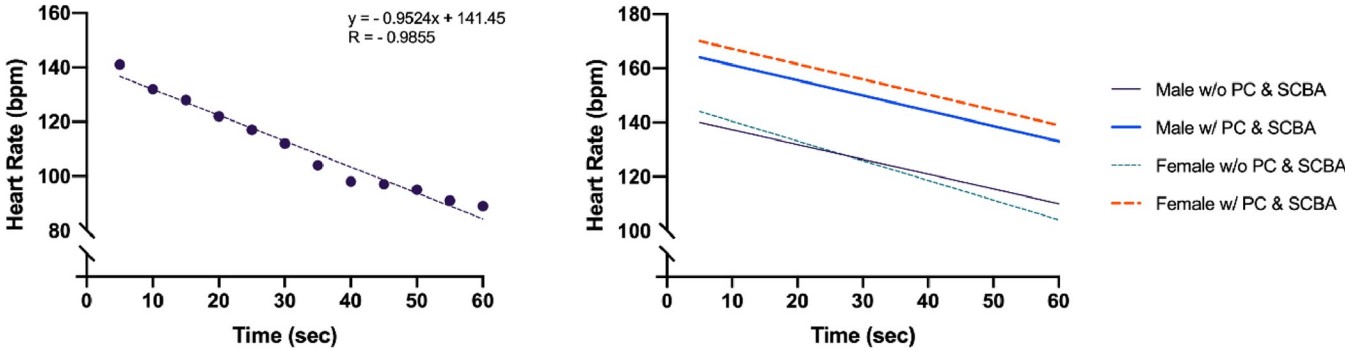

**Fig 3.** Heart rate trend during 1 min post Step test in all subjects (left panel). Trends of heart rate during 1 min post Step test in males and females without equipment (w/o PC & SCBA), and with equipment (w/ PC & SCBA) (right panel).

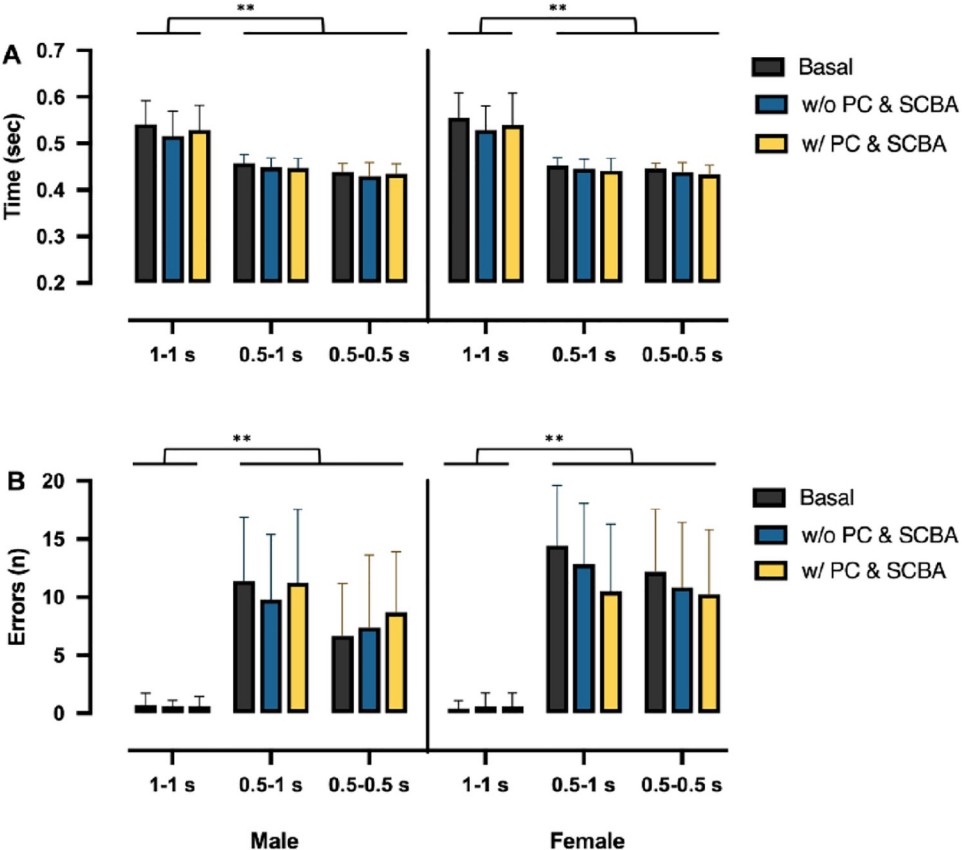

**Fig 4.** Means and standard deviations of the mean reaction time (A) and errors made (B), in males (left) and females (right), for each trail (1–1 s; 0.5–1 s, 0.5–0.5 s), in the three conditions: Basal (before the step test; black columns), without equipment (after step test with only protective clothing, w/o PC & SCBA; blue columns), and with equipment (after step test with protective clothing and Self-Contained Breathing Apparatus, w/ PC & SCBA; yellow columns).

MRT and E were not significantly different. Finally, MRT x trial interaction was extremely significant (P<0.001), and, as expected, slope in T1 was significantly different with respect to T2 and T3 slopes (Fig 5).

## 4. Discussion

The main findings of the present study were: 1) statistical differences in anthropometric and fitness values between male and female Italian firefighter recruits, and 2) no significant effect of overload on the reaction time to visual stimuli, with a significant association between errors and MRT.

Although several studies [40–42] showed physical sex differences in humans, female firefighters complete the same tasks and meet the same fitness standards as men. In addition, firefighter gear is designed for a single-sex, male workforce, and female firefighters have comfort, protection, and mobility restrictions [43].

A previous study of Soteriades et al. [44] showed that firefighters with a BMI of >28.5 kg/m$^2$ had an increased risk of work disability of 70%, while Gendron et al. [26] showing that the prevalence of modifiable CVD risk factors was represented by obesity (12%) and physical inactivity (62%). In addition, obesity may interfere with job performance and increase the risk of injury during firefighting activities [45,46]. In this study, we have found a "normal" average of

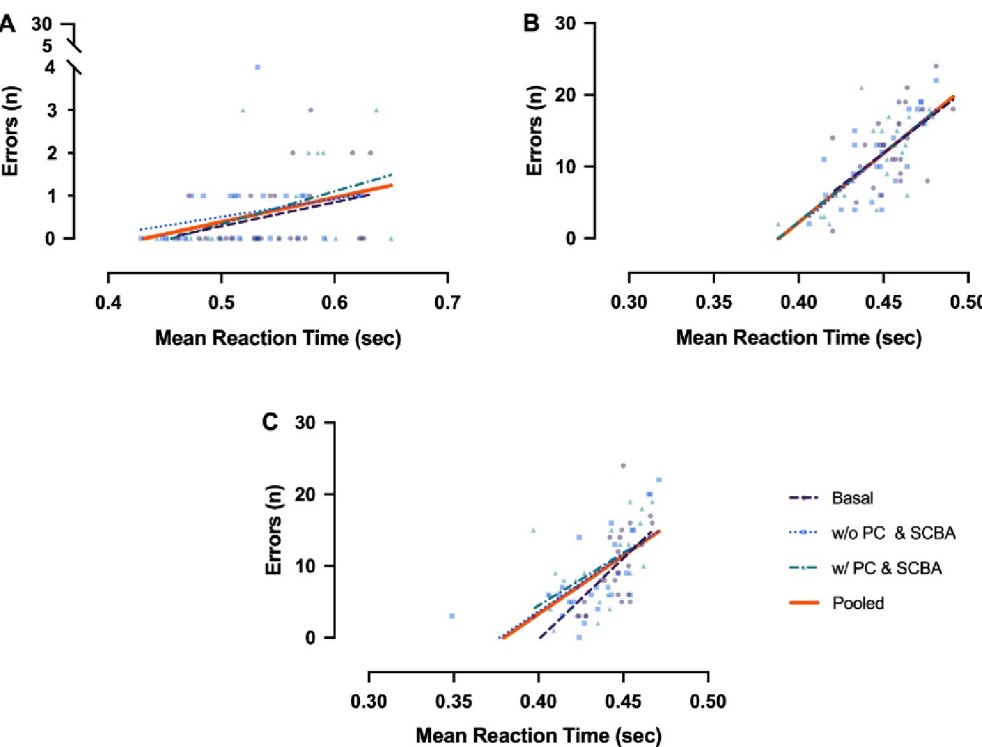

**Fig 5.** Correlations between the mean reaction time and the errors made in basal, without equipment (w/o PC & SCBA), and with equipment (w/ PC & SCBA) conditions in each trial (A: 1–1 s; B: 0.5–1 s, C: 0.5–0.5 s); the pooled lines (in orange) are also represented.

BMI in female ($21.9\pm2.1$ kg/m$^2$) but a lightly "overweight" in male ($25.5\pm2.2$ kg/m$^2$), according to the international BMI classification [47]. Firefighter's BMI values of this study were similar to Gendron et al. [26] for female, but higher than those reported by to Perroni et al. [30] for male for the same age professional firefighters (23.7 and 24.7, respectively). Considering that various studies [48,49] showed a significant weight gained in each year of firefighting career (from 0.5 to 1.5 kg per year), that could represent a gain of 5–15 kg of weight in 10 years of service, it seems appropriate that firefighting departments regularly evaluate the body mass (i.e., BMI, percentage of body fat, and fat-free mass) of their firefighters (recruits and professional) and adopt specific nutritional programs. In addition, firefighting departments should improve the use of more precise measuring instruments for the assessment of obesity than BMI.

Given that RT reflects response efficiency in information processing tasks, as a measure of processing speed, it is essential for firefighters who are constantly exposed to a large number of sensory stimuli (*i.e.*, ambient conditions, colleagues, and other people who may need help). Zhang et al. [50] reported faster response time after a treadmill exercise in a heated chamber, while Hemmatjo et al. [51] showed a faster RT (but with less correct responses) at the end of firefighting activities in a smoke-diving room. Previously, our group [29] found that firefighters made significantly more errors than no firefighters in basal condition, where the RT required was high, underlining a lower level of cognitive function. We hypothesized that the "static" execution of the test did not reproduce the highly variable working conditions on where firefighters usually operate. Compared with our previous study [29], the new results demonstrated low value in both female and male recruits for MRT in all trial, but with a different trend for errors. Despite Morris et al. [52] shown that RT can be impaired when wearing a weighted vest to simulate personal protective equipment while performing a simulated

firefighting task, we didn't find a statistical difference among the three different conditions of execution of RT test in both groups. However, results showed a high correlation between errors and MRT in T2 and T3, and they highlighted a "critical range" between 0.4 and 0.5 s in responses. When the RT in the allowed time (0.5 s) was higher, higher it was the possibility to make an error. When RT is delayed, decision-making ability appeared impaired, and this condition could expose firefighters and potential victims to an increase in the risk of injury. Considering that firefighters perform multiple duties during firefighting activities, which are developed in both day and night shifts (sleeping hours vary greatly depending on the situation of their night work), this test can give us the possibility to evaluate RT in global form. Research is needed to ascertain the neuronal-cognitive component or the muscle-tendon component influence and the different time and type of loads and fatigue on reaction time, to determine future physical fitness testing, and to develop specific firefighting conditioning programs.

It is generally recognized that $VO_{2max}$ represents a measure of cardiovascular health and physical fitness, required to carry out workers' duties safely and effectively. A previous report of Perroni et al. [30] showed a correlation of r = 0.469 between $VO_{2max}$ evaluated by laboratory and field tests, wearing protective garments and SCBA, and a difference of 14% between test results (49.2 and 42.3 $ml^.kg^{-1}.min^{-1}$, respectively). In other studies of the same group [4,53], a high correlation between absolute $VO_{2max}$ values and tests performed with and without PC & SCBA, by firefighters' recruits, had been found. They suggested that $VO_{2max}$ assessed in absolute value could facilitate a comparison between the same firefighters and different subjects during a specific period of physical conditioning. Despite the female showed recommended and better value than male in $VO_{2max}$ measured in weight-related term (45.6 vs. 41.5 $ml^.kg^{-1}.min^{-1}$), in this study we have found higher values in male than in female in $VO_{2max}$ measured in absolute term (3.3 vs. 2.7 $L^.min^{-1}$). Considering only the female group, our results were higher in weight-related terms but lower in absolute term than the study of Kirlin et al. [23] but similar to Li et al. [54] in weight-related term (45.8 $ml^.kg^{-1}.min^{-1}$). Regards to the heart rate responses to exercise, the condition with equipment showed a higher heart rate peak (i.e. higher intercept) and a slower heart rate recovery (i.e. lower slope) respect to the without equipment condition, without gender differences. However, a sex-related difference was found in the slopes of the heart rate recovery, with a significantly faster heart rate decay in females in the without equipment condition, respect to their male colleagues. Similar results in gender-related differences in heart rate recovery were previously reported by other authors [55]. Kirlin et al. [23] found that cardiorespiratory fitness decreases significantly in a sample of female career firefighters ranging in age from 25 to 60 years. In this way, also considering the previous study of Kaminsky et al. [56], which documented a gradual decrease with age (~10% per decade) in cardiorespiratory fitness for both men and women, the finding of our study underscores the need for fire departments to ensure that cardiorespiratory fitness, and its evaluation, continues to be a priority after the intensified training conducted in the new recruits' first year.

In firefighter activities, hands are frequently used for such actions as manual lifting, pushing, pulling, carrying, etc. Grip strength is considered a good index of overall physical strength and a marker of general health, aging, and nutritional status [57,58]. Several studies [59–61] demonstrated that grip strength could indicate full-body strength, neuromuscular activation, and work capacity. Abe et al. [62] recruited 613 adults (50% females) between the age of 20 and 89 years and indicated that age-related decline in handgrip strength is associated with muscle quality. Their results were similar to males (48.2 vs. 48.1 kg) and females (30.2 vs. 31.5 kg) of our study for the same age category. In this study, the results showed lower values respect to those reported by Rostamzadeh et al. [63], which compared maximum handgrip strength between light manual workers and office employees, and found approximately 12.4%

of the difference between groups (59 kg and 51.9 kg, respectively). Both female and male groups showed lower handgrip values than those previously reported by Perroni et al. [4] for novice and volunteer firefighter recruits (49.8 kg and 50.1 kg, respectively). Regarding the sex difference, the handgrips values found in our study were lower than the study of Nazari et al. [64].

Previous studies [65,66] showed that bench-press performances were significantly correlated to specific firefighting performances, while Sheaff et al. [67] found no correlation between endurance bench press performance and firefighters' work capacity. Regarding the female firefighters, Kirlin et al. [23] have shown no significant differences across the age groups in the muscular fitness measured by the maximum number of push-ups, sit-ups, and back endurance. Results of our study about bench press test showed lower estimates and real values than Michaelides et al. [66] and Perroni et al. [68] in male firefighters.

Considering that firefighters might need dynamic strength to perform tasks quickly to help civilians in imminent danger, in this study, we used the CMJ test, which is considered a good measure of functional power of the lower limbs [69,70]. This test has the advantages of a much more natural jumping movement and that the leg muscles reach a higher level of activation and force before they start to shorten [71]. Ryan et al. [72] found that lower limb strength, specifically at fast velocities, was critical for work performance in jobs that were physically demanding. In this study, CMJ values of male were higher than those previously reported by studies of Perroni and colleagues [14,53] for the same age Italian firefighter recruits. Considering that the results of CMJ in the female group in this study showed lower value than male at all different age previously analyzed [14,53], we agree with Rhea et al. [11] that there is a need to include muscle strength and muscle endurance in fitness training of firefighters. Considering that male and female firefighters work side by side in the same strenuous and risky conditions, without limitations about sex, height, weight, or other, these results in recruits at the end of the training course are even more startling.

In this study, we have chosen these tests because they are fast and simple to do, and the movements are similar to those used for the jobs' performance. In addition, they can be used with a large number of firefighters (recruits and career), and to project a prospective follow-up, in each place of National Firefighters Corp, with any age group, in any moment of shift works (day and night, before and after an emergency) without risk injuries. Understandably, our study was subject to several limitations. First, the population was not very large. Second, we only analyzed recruits at the end of the training course and not career firefighters. Third, results depend on the choice of the field test, and the estimate rather than a direct measurement of the parameters of strength and of $VO_{2max}$. Lastly, only one testing session has been performed to obtain measurements related to each participant. Therefore, intra-session ICC cannot be calculated. For these reasons, further research is needed to ascertain the influence of differences in anthropometric, general fitness parameters, and specific responses to stimuli between males and females on firefighters' task and to determine the necessity of developing minimal entry requirements for firefighting recruits and/or specific firefighting conditioning programs.

## 5. Conclusions

Previous studies of different authors indicate that, as a result of physiological sex differences, female firefighters have to reach and maintain a high level of physical ability to perform to the occupational standards expected of firefighters. Considering the main findings found in this study, we emphasize the importance of implementing a specific exercise program for female firefighters to develop adequate levels of strength and endurance to decrease the levels of injury

and increase work capacity. In addition, we strongly suggest to firefighting departments to develop methodologies of training for the RT, which should include specific cognitive tasks (i.e. dynamic firefighters activities), with different quantity of stimulus requests and different time intervals between them. High capabilities of analysis of the situation in a short time (decisional process) could decrease the risk of committing an error during firefighters' emergency activities, which could be fatal.

## Author Contributions

**Conceptualization:** Fabrizio Perroni, Laura Guidetti.

**Data curation:** Fabrizio Perroni, Luca Grandinetti.

**Formal analysis:** Luca Grandinetti, Davide Sisti.

**Investigation:** Ludovica Cardinali, Luca Grandinetti.

**Methodology:** Fabrizio Perroni, Laura Guidetti.

**Project administration:** Fabrizio Perroni.

**Resources:** Lamberto Cignitti, Federico Grugni, Francesco Lunetta.

**Supervision:** Marco Bruno Luigi Rocchi, Vilberto Stocchi.

**Visualization:** Stefano Amatori.

**Writing – original draft:** Fabrizio Perroni, Erica Gobbi, Stefano Amatori.

**Writing – review & editing:** Ludovica Cardinali, Carlo Baldari, Laura Guidetti.

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
