## [Decision Letter · Decision Letter 0]

24 Nov 2020

PONE-D-20-30525

Are there sex differences in physiological parameters and reaction time responses to overload in firefighters?

PLOS ONE

Dear Dr. Guidetti,

Thank you for submitting your manuscript to PLOS ONE. After careful consideration, we feel that it has merit but does not fully meet PLOS ONE’s publication criteria as it currently stands. Therefore, we invite you to submit a revised version of the manuscript that addresses the points raised during the review process.

The manuscript is clear and well written. The topic is interesting and the results could contribute to develop specific training program for male and female firefighters.

I fully endorse the points raised by the reviewer. In addition, I would like to add a few complementary comments of mine which may help increase the clarity of the text.

 Please, consider to include specific hypothesis at the end of introduction; Please, consider to report the ICC values of your measurements (intra-session and inter-day). On the contrary, if you are not able to assess the ICC of the measured variables, you should include this issue in the limitations and reporting (in methods) the ICC values of other studies; Please, consider to limit the use of acronyms; they do not facilitate the reading of the manuscript; Why the estimated VO_2 _max was expressed in absolute terms (l/min)? Movements performed by firefighters are usually executed against the gravity, it could be better if the value would has been expressed per unit of body weight. Please, consider to use the relative value of VO_2 _max (ml/kg/min). This issue is not sufficiently cleared up in the section discussion.

The method used to assess the reaction time should be described in physiological terms considering that it is used to detect the overload. Is it sensitive enough to capture the cognitive changes induced by overload? Can it ascertain the neuronal-cognitive component or the muscle-tendon component (i.e. electromechanical delay)? Please, consider to include these considerations in the discussion in order to expand the reasoning on the variable “reaction time” (i.e. describing the limiting factors of reaction time relatively to the type of test used).  In other words, the reaction time measured in the present study reflects the physiological demands of firefighters in their daily work?    

We look forward to receiving your revised manuscript.

Kind regards,

Riccardo Di Giminiani

Academic Editor

PLOS ONE

Journal Requirements:

2.Thank you for stating the following financial disclosure:

 [he funders had no role in study design, data collection and analysis, decision to publish, or preparation of the manuscript.].

3. Please include your tables as part of your main manuscript and remove the individual files. Please note that supplementary tables (should remain/ be uploaded) as separate "supporting information" files

Reviewers' comments:

Reviewer's Responses to Questions

**Comments to the Author**

1. Is the manuscript technically sound, and do the data support the conclusions?

Reviewer #1: Partly

2. Has the statistical analysis been performed appropriately and rigorously? 

Reviewer #1: Yes

3. Have the authors made all data underlying the findings in their manuscript fully available?

Reviewer #1: Yes

4. Is the manuscript presented in an intelligible fashion and written in standard English?

Reviewer #1: Yes

5. Review Comments to the Author

Reviewer #1: The aim of the current manuscript was to analyze the fitness of male and female Italian firefighters and examine the influence of "overload" on male and female firefighter reaction times.

- while it clear that the number of female firefighters is small compared to males, other than limited data, providing a stronger rationale for examining this population would strengthen the paper. For instance, are female firefighters at a greater risk of physical injury or elevated risk of CVD compared to their male counterparts?

- please include a power calculation.

- If the authors could expand the description of this population of firefighters, it would help the reader. Is this a volunteer department, a paid structure department, a wildland group, etc. How does average age of this population compared to the average age of newly hired recruits in other departments? Additionally, did the 6 months of training provided lead to significant changes in aerobic fitness or strength- have these individuals been habitually active throughout their life or only recently, following the 6 months of training? If only recently active/fit as a result of the 6 months of training, it may influence the interpretation of the findings in the discussion.

-All figures need a legend

-Figure 1 is helpful, but more information can be included on this figure such as a bit more information on T1, T2, T3 (differences between these) and how some tests of performance were assessed (bench press etc.).

-This reviewer did not find Figure 2 necessary or helpful.

-Figure 3 is presented without any mention of the physiological relevance of heart rate recovery. What does recovery of heart rate reflect, why was it measured in this population, and what do any sex differences indicate? This data is presented but not discussed in the discussion and leaves the reader wondering why it was measured and if it is important.

-Figure 4. The authors conclude there is an effect of the trial, but it is unclear if the effect of trial would exist independent of exercise- so a control group (time control) with no exercise and the T1-T3 administered in the same manner would help distinguish if this is a time/exercise effect. Alternatively, in the discussion the authors suggest that RT is lower in FF's compared to other groups, so using a non-FF group as a comparison would be useful in order to understand whether FF's have an improved performance on these tests when "overloaded" compared to a control group- indicating that their training somehow leads to preserved cognitive fx as assessed via RT.

-Table: "estimated" VO2 should be denoted as this was not directly measured. If expressed relative to mass or lean mass, how do the sex differences compare in areas of strength? Given the conditions of the job, I understand that absolute strength and aerobic capacity are more important in terms of FF safety, however, if examining/comparing fitness between sexes, adding a relative strength comparison to the table is relevant.

-Discussion: The abbreviations "MF" "FF" and "E" should be defined - difficult to follow in its current format.

In general, the discussion is very long and can be shortened to address the significant findings.

Conclusion: The authors are suggesting that females "identify with a higher commitment to maintaining their level of physical activity" however, this wasn't assessed. Later, the authors suggest that implementing a specific program for females to develop adequate strength- but where is the evidence that on an absolute scale, their strength is inadequate? if there are current standards that the females are not meeting, these should be presented. Further, do the female FF's need to follow a different program than the males that may need to improve strength? I believe that the female specific recommendations need to be tempered or presented with data that would support the notion that their strength/fitness benchmarks are lacking. Again, referring to the above, do female FF have a greater risk of injury on the job compared to males?

Lastly, there was no difference between sexes or an effect of "overload" in RT, so why would the authors suggest it should be targeted to be improved? Further, is there any evidence in this population that improving RT can reduce injury?

Minor: some grammatical errors.

6. PLOS authors have the option to publish the peer review history of their article (what does this mean?). If published, this will include your full peer review and any attached files.

Reviewer #1: No

---

## [Author Response · Author response to Decision Letter 0]

31 Dec 2020

We thank the Editor and the reviewers for their interesting and useful comments that helped enhancing the quality of our paper. We do hope that, thanks to the reviewers’ comments, we could successfully deal with the requested revisions, rendering the paper better readable. To facilitate the identification of the revisions, we have highlighted in red color all new or modified sentences according to the reviewers

Editor

Thank you for submitting your manuscript to PLOS ONE. After careful consideration, we feel that it has merit but does not fully meet PLOS ONE’s publication criteria as it currently stands. Therefore, we invite you to submit a revised version of the manuscript that addresses the points raised during the review process. The manuscript is clear and well written. The topic is interesting and the results could contribute to develop specific training program for male and female firefighters. I fully endorse the points raised by the reviewer. In addition, I would like to add a few complementary comments of mine which may help increase the clarity of the text.

- Please, consider to include specific hypothesis at the end of introduction;

Answer: According to reviewer’s comments we have changed as follow:

“………………… 6% of male as high CVD risk.

The main goals of the present study were to analyze: 1) the fitness parameters of Italian Firefighters, comparing males and females, 2) the impact of overload, represented by the protective garments, on the reaction time to the stimulus. Despite male and female firefighters work side by side in the same, highly hard, strenuous, and risky conditions, we hypothesized that there are differences in anthropometric and fitness values between male and female Italian firefighter recruits. While much has been published on fitness requirements among male firefighters, data on female firefighters are few.

2. Materials and Methods 

2.1. Participants

Twenty-five healthy recruits ……………………..”

- Please, consider to report the ICC values of your measurements (intra-session and inter-day). On the contrary, if you are not able to assess the ICC of the measured variables, you should include this issue in the limitations and reporting (in methods) the ICC values of other studies;

Answer: Thanks for this comment. Unfortunately, we are not able to report ICC values for intra-session and inter-day measurements. According to the reviewer, we added as a limit of the paper in the relevant section, the following: “Only one testing session has been performed to obtain measurements related to each participant. Therefore, intra-session ICC cannot be calculated.” However, in other articles (Perroni et al., 2015) , the measures were obtained as the same manner as reported in this study.

- Please, consider to limit the use of acronyms; they do not facilitate the reading of the manuscript;

Answer: According to reviewer’s comments we have limited, where it is possible, the use of acronyms along the text

- Why the estimated VO2 max was expressed in absolute terms (l/min)? Movements performed by firefighters are usually executed against the gravity, it could be better if the value would has been expressed per unit of body weight. Please, consider to use the relative value of VO2 max (ml/kg/min). This issue is not sufficiently cleared up in the section discussion.

Answer: We choose to estimate VO2max expressed in absolute terms (l/min) because in our previous research (Perroni et al. 2015), where we investigated the differences of aerobic level in 197 firefighters by Step Test performed with and without fire protective garments and the differences among age, we found a better correlation in L•min-1 than in ml•kg-1•min-1. The research showed that the assessment of VO2max of firefighters in absolute term can be useful tool to evaluate the firefighters' cardiovascular strain than weight related. So, to increase clarity of text we have changed as follow:

“…………, by firefighters’ recruits, had been found. They suggested that VO2max assessed in absolute value could facilitate a comparison between the same firefighters and different subjects during a specific period of physical conditioning. Despite the female showed recommended and better value than male in VO2max measured …………”

- The method used to assess the reaction time should be described in physiological terms considering that it is used to detect the overload. Is it sensitive enough to capture the cognitive changes induced by overload? Can it ascertain the neuronal-cognitive component or the muscle-tendon component (i.e. electromechanical delay)? Please, consider to include these considerations in the discussion in order to expand the reasoning on the variable “reaction time” (i.e. describing the limiting factors of reaction time relatively to the type of test used). In other words, the reaction time measured in the present study reflects the physiological demands of firefighters in their daily work? 

Answer: The same instrument used in this study to evaluate the Reaction Time (FITLIGHT Trainer) was utilized by other research group in students and Airmen (Reigal et al. 2019, Fischer et al. 2015) and it can give us the possibility to evaluate RT only in global form. Previously, our research (Perroni et al 2018) showed significative difference between firefighters and healthy male volunteers in Reaction Time test. In particular, Errors were higher in Firefighters than in control group. Fort his reasons, to increase the knowledge about the influence of overload on Reaction Time in Italian firefighters, we choose to maintain the same procedures. To increase the quality of paper, we have changed as follow:

“Introduction

Firefighting is a hazardous civilian occupation which is responsible for the safety of the public. It is characterized by different and variable (i.e. space, time, duration, breaks, type of the emergency) working conditions, in whose firefighters have to maintain full readiness for 24 hours (divided in shift work) and they are subjected to heavy physical demands and high psychological stress, often working under …..”

“……….execution of the test: step test performed without PC & SCBA and step test performed with PC & SCBA.

2.3.3. Step test

We used a specific field step test compared to non-specific test (i.e. running on a treadmill) because it is a functional activity of the firefighters as climbing stairs [29]. We adopted the YMCA step test [30] consisting of …………………”

“…………both day and night shifts (sleeping hours vary greatly depending on the situation of their night work), this test can give us the possibility to evaluate RT in global form. Research is needed to ascertain the neuronal-cognitive component or the muscle-tendon component influence and the different time and type of loads and fatigue ……………”

Reviewer #1: 

The aim of the current manuscript was to analyze the fitness of male and female Italian firefighters and examine the influence of "overload" on male and female firefighter reaction times.

- while it clear that the number of female firefighters is small compared to males, other than limited data, providing a stronger rationale for examining this population would strengthen the paper. For instance, are female firefighters at a greater risk of physical injury or elevated risk of CVD compared to their male counterparts?

Answer: According to reviewer’s comments we have changed as follow:

“…………….US Reports [24] estimates that female firefighters experienced an average of 1.260 (4%) by 30.290 injuries suffered by all firefighters from 2010 to 2014. Previously, Liao et al. [25] explored the correlation between firefighter injury frequency and duration, demographics, personality and economics, and they found that female firefighters reported 33% more injuries than their male counterparts. According to the 2013 ACSM guidelines, studies of Gendron et al. [26, 27] showed that 11% of female and 34.5% of male firefighters were categorized as having moderate cardiovascular disease (CVD) risk, while 65% of female and 43.6% of male as high CVD risk.

The main goals of the ………………….”

- please include a power calculation.

Answer: Power analysis was performed considering ANOVA models with repeated measured, within factor. In a two-way ANOVA analysis (gender was between factor, MRT was three levels within factor), a sample of 24 subjects, achieves a power (1-β) of 0.80. The group was balanced and a mean correlation index 0f 0.4 among repeated measures was estimated. The effect size f, which is calculated using f = (σm / σ), is equal to 0.3. This power assumes an F test is used with a significance level (α) of 0.05.

- If the authors could expand the description of this population of firefighters, it would help the reader. Is this a volunteer department, a paid structure department, a wildland group, etc. How does average age of this population compared to the average age of newly hired recruits in other departments? Additionally, did the 6 months of training provided lead to significant changes in aerobic fitness or strength- have these individuals been habitually active throughout their life or only recently, following the 6 months of training? If only recently active/fit as a result of the 6 months of training, it may influence the interpretation of the findings in the discussion.

Answer: According to reviewer’s comments we have changed as follow:

“……………… requirements among male firefighters, data on female firefighters are few. 

2. Materials and Methods 

2.1. Participants

Twenty-five healthy recruits of National Italian Firefighters Corp (Age: 37 ± 4 years; Weight: 69.1 ± 12.2 kg; Height: 169 ± 7 cm; BMI: 23.7 ± 2.8 kg/m2) consisting of 12 female firefighters (FF) and 13 male firefighters (MF) were recruited to participate in this study. To avoid possible statistical differences derived by different activity levels and dietary habits, they were engaged at the end of the residential Italian Fire Fighter Corp training course, organized by Minister of Internal Affair in its public structures to became professional firefighters. The Fire Fighter Corp training course consisted of 6 months (8 h/day, 5 days/week) of firefighting education and training, which included theoretical and practical firefighting skills (i.e., emergency medical procedures, fire apparatus operations, fire prevention, and communication practices) and strength and conditioning training. Before the course, the Italian Fire Fighting Corp selected recruits according to questionnaires, interviews, and fitness tests.

All subjects signed a consent form. ……..”

- All figures need a legend

Answer: According to the reviewer’s comment, all the figures have a legend. Figures’ captions have been slightly modified in order to improve clarity.

- Figure 1 is helpful, but more information can be included on this figure such as a bit more information on T1, T2, T3 (differences between these) and how some tests of performance were assessed (bench press etc.).

Answer: According to the reviewer’s comment, Figure 1 has been modified, in order to make it more informative and easier to understand.

- This reviewer did not find Figure 2 necessary or helpful.

Answer: We thank the reviewer for his suggestion but, for a previous (Perroni et al. 2015) and possible future comparison with other groups, we prefer to maintain the Figure 2. We think that it can give more information to reader and to increase the clarity of the execution of test, as it shows how the Fitlights were positioned and allow the reproducibility of the test.

- Figure 3 is presented without any mention of the physiological relevance of heart rate recovery. What does recovery of heart rate reflect, why was it measured in this population, and what do any sex differences indicate? This data is presented but not discussed in the discussion and leaves the reader wondering why it was measured and if it is important.

Answer: We thank the reviewer for this comment. According to his comments we have added a relevant part in the Discussion section, as follows: 

“Considering only the female group, our results were higher in weight-related terms but lower in absolute term than the study of Kirlin et al. [23] but similar to Li et al. [53] in weight-related term (45.8 ml.kg-1.min-1). Regards to the heart rate responses to exercise, the condition with equipment showed a higher heart rate peak (i.e. higher intercept) and a slower heart rate recovery (i.e. lower slope) respect to the without equipment condition, without gender differences. However, a sex-related difference was found in the slopes of the heart rate recovery, with a significantly faster heart rate decay in females in the without equipment condition, respect to their male colleagues. Similar results in gender-related differences in heart rate recovery were previously reported by other authors [54]. Kirlin et al. [23] found that cardiorespiratory fitness decreases significantly in a sample of female career firefighters ranging in age from 25 to 60 years.”

- Figure 4. The authors conclude there is an effect of the trial, but it is unclear if the effect of trial would exist independent of exercise- so a control group (time control) with no exercise and the T1-T3 administered in the same manner would help distinguish if this is a time/exercise effect. Alternatively, in the discussion the authors suggest that RT is lower in FF's compared to other groups, so using a non-FF group as a comparison would be useful in order to understand whether FF's have an improved performance on these tests when "overloaded" compared to a control group- indicating that their training somehow leads to preserved cognitive fx as assessed via RT.

Answer: We thank the reviewer for this comment. If we correctly understood what reviewer means, he asks about a control group without exercise. However, in the Basal condition (dark grey columns) participants performed the reaction time test before any type of exercise, so this could be considered as a control condition (no exercise). Being the results of the trials (T1 vs T2 vs T3) also different in the basal condition, we can say that the effect of the trial exists independently from exercise. As already reported above, a previous research of our group (Perroni et al., 2018) showed a significative difference between firefighters and healthy male volunteers in Reaction Time test, with the same experimental procedures.

- Table: "estimated" VO2 should be denoted as this was not directly measured. If expressed relative to mass or lean mass, how do the sex differences compare in areas of strength? Given the conditions of the job, I understand that absolute strength and aerobic capacity are more important in terms of FF safety, however, if examining/comparing fitness between sexes, adding a relative strength comparison to the table is relevant.

Answer: Considering that male and female firefighters work side by side in the same, highly hard, strenuous, and risky conditions, we think that it is more important the evaluation of absolute strength than relative strength. 

- Discussion: The abbreviations "MF" "FF" and "E" should be defined - difficult to follow in its current format. In general, the discussion is very long and can be shortened to address the significant findings.

Answer: To increase the clarity of paper, according to reviewer’s comments we have changed MF with male, FF with female, and E with errors. In addition, we deleted some parts to reduce the discussion and to highlight the significant findings. 

- Conclusion: The authors are suggesting that females "identify with a higher commitment to maintaining their level of physical activity" however, this wasn't assessed. Later, the authors suggest that implementing a specific program for females to develop adequate strength- but where is the evidence that on an absolute scale, their strength is inadequate? if there are current standards that the females are not meeting, these should be presented. Further, do the female FF's need to follow a different program than the males that may need to improve strength? I believe that the female specific recommendations need to be tempered or presented with data that would support the notion that their strength/fitness benchmarks are lacking. Again, referring to the above, do female FF have a greater risk of injury on the job compared to males? Lastly, there was no difference between sexes or an effect of "overload" in RT, so why would the authors suggest it should be targeted to be improved? Further, is there any evidence in this population that improving RT can reduce injury?

Answer: As wrote in Introduction section, previous research of different authors highlighted that the job performance of firefighters is strenuous and heavy. Considering that in Italian National Firefighters Corp there aren’t different role in the manage of job performance of firefighters (all have to do any activities required by emergencies), the strength tests in this article are presented in absolute term and no in relative term. So, female starts with a disadvantage compared to male and they have to reach and maintain high level of physical ability to perform to the occupational standards expected of firefighters. In this study, we analysed “static” execution and we suggest to study more dynamic and “functional” activities. RT is important factor in “decisional process” and its velocity of analysis is useful to be efficient in emergencies as to exit by “critical situation”. To increase the clarity of paper, according to reviewer’s comments we have changed as follow:

“………firefighting conditioning programs.

5. Conclusions

Previous studies of different authors indicate that, as a result of physiological sex differences, female firefighters have to reach and maintain a high level of physical ability to perform to the occupational standards expected of firefighters…..”

“…….. an average of 1.260 (4%) by 30.290 injuries suffered by all firefighters from 2010 to 2014. Previously, Liao et al. [25] explored the correlation between firefighter injury frequency and duration, demographics, personality and economics, and they found that female firefighters reported 33% more injuries than their male counterparts. According to the 2013 ACSM guidelines, studies of Gendron et al. [26, 27] showed that 11% of female and 34.5% of male firefighters were categorized as having moderate cardiovascular disease (CVD) risk, while 65% of female and 43.6% of male as high CVD risk.

The main goals of the present ……..”

“…………..for the RT, which should include specific cognitive tasks (i.e. dynamic firefighters activities), with different ………”

“……..them. High capabilities of analysis of the situation in a short time (decisional process) could decrease the risk……………...”

- Minor: some grammatical errors.

Answer: According to reviewer’s comments we have done a check in all text.

---

## [Decision Letter · Decision Letter 1]

25 Jan 2021

PONE-D-20-30525R1

Are there sex differences in physiological parameters and reaction time responses to overload in firefighters?

PLOS ONE

Dear Dr.Guidetti,

Thank you for submitting your manuscript to PLOS ONE. After careful consideration, we feel that it has merit but does not fully meet PLOS ONE’s publication criteria as it currently stands. Therefore, we invite you to submit a revised version of the manuscript that addresses the points raised during the review process.

We look forward to receiving your revised manuscript.

Kind regards,

Riccardo Di Giminiani

Academic Editor

PLOS ONE

Reviewers' comments:

Reviewer's Responses to Questions

**Comments to the Author**

1. If the authors have adequately addressed your comments raised in a previous round of review and you feel that this manuscript is now acceptable for publication, you may indicate that here to bypass the “Comments to the Author” section, enter your conflict of interest statement in the “Confidential to Editor” section, and submit your "Accept" recommendation.

Reviewer #1: (No Response)

2. Is the manuscript technically sound, and do the data support the conclusions?

Reviewer #1: Partly

3. Has the statistical analysis been performed appropriately and rigorously? 

Reviewer #1: Yes

4. Have the authors made all data underlying the findings in their manuscript fully available?

Reviewer #1: No

5. Is the manuscript presented in an intelligible fashion and written in standard English?

Reviewer #1: (No Response)

6. Review Comments to the Author

Reviewer #1: Thank you for your response to my concerns. Regarding the reaction time data, this reviewer strongly feels that the authors need control data. I understand that the reaction time baseline value was taken prior to exercise, but what happens to the reaction time value if you repeat it three times without exercise in this same population? How variable is this measure? What happens to the reaction time if you were to repeat this test in a non-firefighter population? There needs to be additional control data to significantly strengthen this manuscript.

Further, while the amended manuscript includes more of a rational for studying female firefighters (injury risk), the physiological relevance of a sex difference related to the slope of the line of the heart rate reserve, is unclear. What does a difference in slope indicate and how does this relate to the primary research question or risk of injury? Further, was the peak heart rate greater in females vs. males?

7. PLOS authors have the option to publish the peer review history of their article (what does this mean?). If published, this will include your full peer review and any attached files.

Reviewer #1: No

---

## [Author Response · Author response to Decision Letter 1]

8 Feb 2021

Reviewer #1: Thank you for your response to my concerns. Regarding the reaction time data, this reviewer strongly feels that the authors need control data. I understand that the reaction time baseline value was taken prior to exercise, but what happens to the reaction time value if you repeat it three times without exercise in this same population? How variable is this measure? What happens to the reaction time if you were to repeat this test in a non-firefighter population? There needs to be additional control data to significantly strengthen this manuscript.

Answer: To evaluate the RT, previous researches used FITLIGHT TrainerTM . For example, Zwierko et al. (2014) showed that non-athletes had longer RTs than athletes, Fischer et al. (2015) used this instrument in the United States Air Force, and Zurek et al. (2015) and Genovese et al. (2020) investigated the simple and complex RTs of athletes players who had undergone knee surgery and a rehabilitation program to assess their recovery. In addition, Reigal et al. (2019) studied the reliability of the FITLIGHT TrainerTM using the intraclass correlation index (IC), the standard error of measurement (SEM), and the minimal difference (MD). The results showed ICC2;1 = 0.92, SEM = 39.87, and MD = 110.51 ms for the simple task. For the complex task, they found ICC2;1 = 0.85, SEM = 63.50, and MD = 176.00 ms, which can be considered as adequate reliability indices.

To increase the clarity of text we changed as follow: 

“…………………

Reaction Time Test

The Fitlight Trainer ® (FitLight Sports Corp, Ontario, Canada) was used to assed the RT to visual stimulus. It is a wireless reaction system, comprised of eight LED lights controlled by a tablet computer; a sensor reacts to proximity or touch and switches off the light. Previous study of Reigal et al. (2019) found adequate reliability indices of the Fitlight Trainer ® for simple (ICC2;1 = 0.92, SEM = 39.87, and MD = 110.51 ms) and complex (ICC2;1 = 0.85, SEM = 63.50, and MD = 176.00 ms) task. Participants, standing upright with arms along the hips, at about 30 cm from the wall, centred to………….”

“Previously, our group [28] found that firefighters made significantly more errors than no firefighters in basal condition, where the RT required was high, underlining a lower level of cognitive function. We hypothesized that the “static” execution of the test did not reproduce the highly variable working conditions on where firefighters usually operate. Compared with our previous study [28], the new results demonstrated low value in both female and male recruits for MRT in all trial, but with a different trend for errors.”

Further, while the amended manuscript includes more of a rational for studying female firefighters (injury risk), the physiological relevance of a sex difference related to the slope of the line of the heart rate reserve, is unclear. What does a difference in slope indicate and how does this relate to the primary research question or risk of injury? Further, was the peak heart rate greater in females vs. males?

Answer: As we wrote in revised text the main goals of the present study were to analyse: 1) the fitness parameters of Italian Firefighters, comparing males and females, 2) the impact of overload, represented by the protective garments, on the reaction time to the stimulus. We didn’t talk about heart rate reserve and about injury. We analysed the sex differences in function of job performance considering that lower cardiovascular fitness may induce higher fatigue that may reduce alertness and result in poor job performance and could also lead to an injury. Heart rate recovery is commonly used as an indicator of cardiovascular fitness (Shetler et al., 2001). So, high decrease of Heart Rate in short time after hard exercise is an index of good fitness.

To increase the clarity of text we changed as follow: 

“……………. anthropometric and fitness values between male and female Italian firefighter recruits. This study could give useful information to develop appropriate conditioning programmes aimed to reach an optimal job performance of firefighters. In fact, while much has been published…….”

---

## [Decision Letter · Decision Letter 2]

22 Mar 2021

Are there sex differences in physiological parameters and reaction time responses to overload in firefighters?

PONE-D-20-30525R2

Dear Dr. Guidetti,

We’re pleased to inform you that your manuscript has been judged scientifically suitable for publication and will be formally accepted for publication once it meets all outstanding technical requirements.

Kind regards,

Riccardo Di Giminiani

Academic Editor

PLOS ONE

Additional Editor Comments (optional):

Reviewers' comments:

Reviewer's Responses to Questions

**Comments to the Author**

1. If the authors have adequately addressed your comments raised in a previous round of review and you feel that this manuscript is now acceptable for publication, you may indicate that here to bypass the “Comments to the Author” section, enter your conflict of interest statement in the “Confidential to Editor” section, and submit your "Accept" recommendation.

Reviewer #1: All comments have been addressed

2. Is the manuscript technically sound, and do the data support the conclusions?

Reviewer #1: Yes

3. Has the statistical analysis been performed appropriately and rigorously? 

Reviewer #1: Yes

4. Have the authors made all data underlying the findings in their manuscript fully available?

Reviewer #1: Yes

5. Is the manuscript presented in an intelligible fashion and written in standard English?

Reviewer #1: Yes

6. Review Comments to the Author

Reviewer #1: (No Response)

7. PLOS authors have the option to publish the peer review history of their article (what does this mean?). If published, this will include your full peer review and any attached files.

Reviewer #1: No

---

## [Editor Report · Acceptance letter]

31 Mar 2021

PONE-D-20-30525R2 

Are there sex differences in physiological parameters and reaction time responses to overload in firefighters? 

Dear Dr. Guidetti:

I'm pleased to inform you that your manuscript has been deemed suitable for publication in PLOS ONE. Congratulations! Your manuscript is now with our production department. 

Kind regards, 

on behalf of

Prof. Riccardo Di Giminiani 

Academic Editor

PLOS ONE